# Religion upon the Mountains: From Christianisation to Social Actions against Summit Crosses in Italy

Giovanna Rech 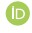

Department of Human Sciences, University of Verona, 37129 Verona, Italy; giovanna.rech@univr.it

**Abstract:** In Italy, the debate regarding the presence of crosses and crucifixes in public places is long-standing and involves their detractors, supporters and defenders. Over time, these conflicting positions have gained media resonance, becoming a sociopolitical controversy that has led to lawsuits at various levels, including the European Court of Human Rights. In the social sphere, the issue has oscillated between the recognition of the universal value of religious symbols and advocacy for secularism, even in open spaces such as mountaintops. During the last few decades, several initiatives have been undertaken in the Italian Alps, driven by ecological concerns and opposition to the presence of crosses on the mountains. These initiatives have resulted in collective actions against the positioning and erection of crosses, and there have even been attempts to diversify the Italian peaks. By providing a historical overview of the Christianisation of Italian mountaintops and focusing on the mobilisation against the presence of crosses, this article contributes to the understanding of the role of such symbols in Italian public opinion, which is intertwined with the vitality of the Catholic Church and the sociopolitical implications of these initiatives. The research questions will investigate the process of legitimisation and delegitimisation of Christian symbols. The cross on the mountaintop serves as an example of culturalised religion, where this cultural object can become a "passive religious symbol," polarising claims for the defence of the natural environment and the sustainability of religion in the mountains.

**Keywords:** summit crosses; mountain; social actions; Christian symbols; Italy

## 1. Introduction

This article illustrates the conflict surrounding summit crosses in Italian public opinion among individuals interested in mountains, mountaineering and nature conservation over the past two decades. Mountains hold symbolic representations, primordial myths and legends. In modernity, European mountains have become territories of symbolic and political conquest, sporting challenges and scientific experimentation. Even today, mountains continue to be social spaces that nourish both collective and individual imagination, with a functional and symbolic interdependence between territories.

Italian mountains have been subjected to colonisation at the socioeconomic policy level (Barbera 2020) and have been exploited for recreational and sporting purposes for well over a century (Battilani 2009; Leonardi 2017). Social representations involving spiritual and religious aspects persist to this day (Kakalis and Goetsch 2018; Gardner 2002). These representations are influenced by two contrasting processes rooted in global media culture. On the one hand, there is the postmodern perception of nature that oscillates between exalting wilderness (Bell 1993; Mathieu 2022) and its enchantment and Disneyfication (Camorrino 2020). On the other hand, Western societies' religious pluralism, which arrived later in Italy, introduced mechanisms to legitimise religious diversity in various social contexts (Giordan and Pace 2014), including ecosystems and mountains (Bernbaum 2022).

This article identifies a distinctive configuration that has emerged within the public discourse concerning the mountain environment and the objects placed there, forming a dynamic relationship between the religious and secular realms (Knott 2013). The research

questions investigate the process of legitimising and delegitimising Christian symbols that have characterised many Italian peaks since the second half of the 19th century. Specifically, the focus is on the cross on the mountaintop, treated as a 'cultural object' (Griswold 2013), testing the hypothesis that it can be considered a 'passive religious symbol' today (Breskaya et al. 2022), on the theoretical basis of Simmelian sociology. In the Italian case, the contestation of the cross polarises claims for the defence of the natural environment and the mountain as a public space. Central to this reflection is the discussion of sustainability in relation to religion (Silvern and Davis 2021) or the unsustainability of religious attractions and religiosity in the mountains (Rech 2022). Thus, this article will contribute to understanding the role of such symbols in Italian public opinion between a presumed vitality of the Catholic Church, which has become a promoter of ecological claims (Turina 2022), and the sociopolitical implications underlying such initiatives. The article is set in the context of the pluralisation of Catholicism (Bova 2017) and the culturalisation of the symbols of the Catholic religion (Astor and Mayrl 2020) relating to both the majority and minorities (Beaman 2012).

After briefly discussing materials and methods, the first part of this article frames the study through a comprehensive analysis of mountains, considering the legal regulation of landscape aesthetics, the related environmental and spiritual ethics, and the status of religious symbols in the public sphere from the perspective of cultural studies on religion. The second part is devoted to the ascending branch of the parabola of the history of mountain crosses; that is, the process of the modern Christianisation of mountain summits. The third part examines the descending branch of this parabola, with the contested and opposed demand for religious freedom in the mountains, disapproving the idea of a Catholic identity shared by the majority of the population. The ensuing discussion articulates what is at stake in today's movement of contestation of crosses in the Italian mountains, starting from a Simmelian analysis of the cross and the mechanisms of religious culturisation enacted by this movement.

## 2. Materials and Methods

The article is based on qualitative research which, through different research interests, collected various ethnographic materials: natural and publicly available documents, non-directive interviews, informal conversations and netnographic observations. These materials relate to the extensive and nearly twenty-year-long movement of opinion and social action concerning crosses (press, social media, and blogs), as well as public demonstrations and collective action mobilising against the presence or erection of crosses in the mountains of the Alpine arc.

The mobilisation of opinion is analysed as a 'discursive opportunity' provided by the media (della Porta and Pavan 2018) and certain online petition platforms capable of generating social movement (Harrison et al. 2022). Although the impact on the transformation of the social order remains modest, it holds significant implications for understanding how the cross has become symbolically appropriated as a cultural object within public opinion.

## 3. Literature Review: Mountains and Religious Symbols

### 3.1. The Mountain as a Social Place: Landscape Regulation

The mountain is not simply nature or a landscape; it is, in fact, a spatial and geographical entity and requires sociological understanding (Freudenburg et al. 1995). Between the 19th and 20th centuries, mountains served as laboratories for experiments, measurements, and scientific observations (Cuaz 2009). As Simmel (1993) himself observes, mountains are defined in contrast to human forms, resulting in a certain dependence on urban environments. Much of the literature on mountains presents a dichotomy of plain–mountain or centre–periphery, highlighting the functional and political interdependence between highlands and urban centres (Dematteis et al. 2017). According to Simpson (2019), even the most recent studies focus less on the intrinsic form or content of mountains and more

on how they "contrast with surrounding areas in ways that include, but almost always go beyond, vertical difference" (p. 554).

During the 19th century, the mountain evolved into both an anthropic and natural space to be protected and preserved, characterised by climatic and environmental features that evoke picturesque beauty, as exemplified by the Dolomites (Bainbridge 2018). Moreover, it represents an imaginary realm of tranquillity, enjoyment, and rest (De Rossi 2014). With the progressive democratisation of leisure time and the widespread consumption associated with it, the Alpine arc became an ideal destination for recreation, sports, and competitions (Boyer 2004). According to the Italian sociologist, Savelli (2012), the conquest and utilisation of the mountains for tourism purposes embodies the self-direction inherent in the rising European bourgeois mentality at the time. Savelli states: "The conquests of the mountains combine the respect and admiration for nature, typical of the Romantic movement, with the involvement of the whole person" (Savelli 2012, p. 123).

In appreciating the landscape, the grandeur with which the mountain, or rather the Alps—as they were antonomastically referred to in the 19th-century tradition—present themselves to human observation (Simmel 1993), implies a predominance of logic, rationality and urban decision-making over the mountainous environment. Despite the various ways in which the mountain holds primary importance as an ideal and cosmic centre in the history of religions (Eliade 1952), Western religious thought has only relatively recently recognised its aesthetic and spiritual value (Nicolson 1997). Recent censuses, conducted with scientific enthusiasm (Fraschia 1997; Mathis 2002; Löwer 2019; Millesimi 2022) or from a spiritually engaged perspective (Jeanneret 2018), have investigated European summit crosses from different angles. These works explore a topic with multiple scientific implications, prompting reflection on moral, spiritual, naturalistic, political, historical and even theological and pastoral matters.

Italy has two main mountain systems (the Alps and the Apennines) with profoundly different natural and cultural characteristics. However, the historical and political importance of the Alps is primarily linked to their centuries-old role as a hinge of civilisations and economic exchange. In the second half of the 20th century, European unification and ecologisation largely determined politics, bringing tension between internal and external claims for the use and non-use of Alpine territories (Mathieu 2022).

In Italy, the legal regulation of mountains concerning the aesthetic value of the landscape, is relatively recent. Although this was a 'constitutional commitment' in the 1940s, it finally took shape with the comprehensive law on cultural heritage and landscape (Codice dei beni culturali e del paesaggio, henceforth CBCP), which became effective in 2004 (Losavio 2017). However, within the current Italian legal framework, new cross installations are considered constructions that require legal permits at the local level (usually municipalities), with the binding ruling lying with the regional authorities through the Superintendency of Archaeology, Fine Arts and Landscape if the work is to take place in a protected area with a landscape bond (CBCP 2004). If the installation is planned in an environmentally protected area, depending on the degree of protection (local, regional, national or part of the Natura 2000 European network), the project requires an environmental impact assessment established by experts.

### 3.2. Environment, Religions and Spiritual Ethics

The interaction between human societies, the environment and symbolic systems has coexisted with humankind. However, it is the global awareness of ecological crises that has accelerated reflection on the relationship between ecology, religions and spirituality. This topic has been at the forefront since the highly influential article by White (1967), whose thesis concerned the role of Western religious traditions—mainly Christianity—and their impact on the conception of man and nature through the development of science and technology. In recent decades, as anthropogenic damage to the environment becomes increasingly evident, ecological issues have permeated many fields of knowledge. A branch of religious studies has been devoted to the environmental humanities, investigating the



relationship between religion and ecology from historical, sociological and theological perspectives (Jenkins 2017). Environmental theology is older (Doughty 1981) than the recent and growing scholarship (Northcott and Scott 2014; Jenkins 2008). Taking a narrower perspective, the Catholic social doctrine of the Church still centres on humanism and solidarity among humans (Pontifical Council for Justice and Peace 2004) rather than on nature, whose vocation continues to be supernatural as created by God (Turina 2013).

In more contemporary times, mountain representations, on par with other ecosystems, also encompass the spiritual and transcendent realm (Rime 2021; Bernbaum 2006; Brunet et al. 2005; Gallingani 2002; Roux 1999). Their significance goes beyond the humanities and social sciences, and contributes to the implementation of global and local nature protection policies. The spiritual dimension of nature protection and the potential for values-based education that draws on spirituality and religiosity for safeguarding nature and biodiversity, have been discussed in United Nations advisory institutions such as the International Union for Conservation of Nature (IUCN) (Mallarach and Papayannis 2007; Pungetti 2012). Religions, therefore, play an active role in society and environmental conservation such as in Öhlmann and Swart's (2022) contribution to the the last report of the Joint Learning Initiative on Faith and Local Communities (2022).

However, it should be noted that this text does not delve into a purely symbolic nor theological understanding of the mountain and its historical cosmogony (Roux 1999), as the focus is on protests against crosses, which belong to an anthropised and recreational present rather than a mythical past. The symbolic conflicts examined here imply values related to the relationship between individuals, society and the environment, also invoking aesthetic considerations. They can also be seen as the expression of social relations with nature (Berghöfer et al. 2022). Furthermore, the mountain is not solely perceived as a natural space, but also as a social and cognitive realm where symbolic representations and religious beliefs can be expressed and where social life develops with elements of integration or conflict (Gunzburg et al. 2021). Mountain peaks are frequented by mountaineers and walkers and are subject to recreational and ostensive social uses ranging from mountaineering to tourism, to the performative practices of selfies. For this reason, they should now be understood as social and public spaces where personal and collective identities are disseminated and amplified through various media platforms.

*3.3. Public Sphere, Religious Symbols and Cultural Objects*

The role of religious symbols in public space has been a long-debated topic in the legal sphere (Di Cosimo 2020; Palese and Catenazzi 2013; Blanke 2012), particularly concerning the display of crucifixes in classrooms, which originated approximately thirty years ago in Italy. Within the Italian context, this issue has raised questions among historians (Luzzatto 2011) and anthropologists (Gallini 2007). Over time, the debate has become a subject of controversy that has polarised political affiliations in Italy (Ozzano and Giorgi 2013). Lawsuits have been filed addressing religious freedom, leading to several rulings at various judicial levels, including the European Court of Human Rights. The most recent ruling of the Italian Court of Cassation is from 2021, bringing the topic back to the present by exploring the multiple meanings that the Christian symbol can acquire in religious and public contexts (Breskaya et al. 2022). As in other national contexts, this leads to questions about the culturalisation of the Catholic religion in relation to the sphere of law (Burchardt 2020; Joppke 2018). According to Joppke (2018), there is an "effective source of culturalizing majority religion in Europe, which is state actors, often in the form of high courts" (p. 238). For democratic and liberal states, the dilemma is real because conflicts about symbolic objects such as the crucifix could disturb neutrality and the "regulative idea" of separation (Joppke 2018, p. 244). Moreover, this debate reflects the specific configuration that secularism has assumed within the Italian context (Zamagni and Guarnieri 2009; Nicoletti 2005; Rusconi 2000).

In the context of cultural studies on religion, which underpins this research, the question refers not only to the Durkheimian dialectic of the sacred and the profane, but

also to the experience of the external world as a landscape within modernity (Simmel 2006). Furthermore, it benefits from the spatial turn that has characterised the study of space, place and religion in recent decades, as illustrated in sociology by Knott (2010), and earlier by anthropologists such as J.Z. Smith (1978). However, the specificity of the sacred or religious nature of the objects under consideration is part of an increasingly evident phenomenon in the West, namely, the culturalisation of religion, as identified by Astor and Mayrl (2020). Thus, religious identities, symbols and institutional forms transcend purely confessional principles and adapt to the cultural sphere. This study on summit crosses explicitly contributes to Astor and Mayrl's (2020) research agenda by examining the conditions under which the presence of religious artefacts influences religious sensibilities as well as non-religious or anti-religious sensibilities (Astor and Mayrl 2020).

Coming to the contested material object, namely, the cross erected on mountain tops, it is mostly devoid of human presence (which would make it a crucifix), and is understood here not so much in its purely symbolic sign dimension but as an external objectification of messages. The symbolic sign dimension is obviously relevant, as eloquently described in Guénon's (1957) dense essay published in 1931. However, considering the peculiarity of the analysed material—-that is, a discursive capital on the contested presence and erection of crosses on Italian peaks—-it aligns better with Griswold's (2013) definition of a cultural object. It can be seen as "a shared meaning embodied in form" or a "socially significant expression that is audible, visible, or tangible or that can be articulated" (Griswold 2013, p. 11). Viewing the cross as a cultural object firstly allows for the incorporation of its symbolic character into the tangible artefact for those who reach it and makes it visible to those who appreciate its image. Thus, it captures a part of the broader system to which it refers, namely, the religious universe and its relationship with the sacred (Vidal 1989). Secondly, as a cultural object, it can influence social or individual behaviour through empirically detectable dynamics, such as mobilisation in the media, collective actions and public events that thematise it. Of primary importance for its analytical implications is the status of cultural objects in light of Simmelian sociology and its analysis of space qualities (Simmel 1998). This perspective allows us to focus on the cross in its "tragic" object–subject dualism and the conflict arising from the insertion of the human into natural reality, a conflict that Simmel (2012) considers permanent.

The conflict between the two cultural and subjective dimensions is aptly illustrated when considering, or not, the cross as a "passive religious symbol" in the sense accepted by Breskaya et al. (2022) considering the jurisprudence of the European Court of Human Rights such as in Beaman's (2012) analysis. "According to the European Court's jurisprudence, a religious symbol or practice can be qualified as "passive" if it is incapable of producing any chilling effect on the freedom of religion and belief of those exposed to it" (Breskaya et al. 2022, p. 16). While the purely legal sociological implications are beyond the scope of this study, it should be noted that conflicts have also occurred in this area, and calls for the secularity of the state are an evident manifestation of this.

The conflict engendered by the presence or erection of new or renewed crosses in the Italian public sphere will be discussed, firstly, through a historical contextualisation of their presence and, secondly, through an analysis of the stakes involved in the debate between detractors and supporters of the cross on Italian peaks since the beginning of the 2000s.

## 4. Crosses in the Mountains: A Christianisation of the Peaks

The practice of installing crosses on mountain summits in the Alps has been widely practised in recent centuries, and while providing a comprehensive history of it is beyond the scope of this article, it is useful to understand the origins of this phenomenon (Acreman 2019; Grillmayer 2022). Historians who contributed to the 2006 special issue of the journal, *Mountain Research and Development*, considered this practice as the "sacralisation of mountains," which gained prominence in the modern era. Mathieu (2006) argues that starting from the 16th century, there was an expansion of sacred mountain landscapes, specifically the topography of the sacred that accompanied travellers and, more importantly,

pilgrims through passes or in ancient xenodochia, but not necessarily on mountain peaks. It was around the turn of the 19th century that the erection of the first Catholic crosses and, much later, Protestant crosses on the peaks of the Alpine region can be historically documented (Anker 2012). In this sense, the European mountain was not only "sacralised" but also "Christianised" through a symbolic colonisation by certain Catholic and Christian mountaineers.

This phenomenon is particularly evident in the case of Italy. According to the historian, Cuaz (2006, 2009), the educational significance of mountaineering was particularly important in Italian Catholic circles. Indeed, in the second half of the 19th century, mountaineering in Italy took on a distinct tone for the Catholic Church, with organisations such as the Giovane Montagna. Mountains and mountaineering practice became a means of indirect social control, which clerics and associations employed as a discipline and a form of moral guidance during leisure time (Cuaz 2006, p. 359). From the 1890s onwards, religious artefacts and ritual practices such as high-altitude masses became quantitatively significant, representing not "isolated initiatives but a project for the Christianisation of summits and the establishment of a Catholic alpinism in clear opposition to the nationalist tendencies of secular mountaineering" (Cuaz 2009, p. 62).

An important event during this period is related to the Jubilee of 1900, which became an opportunity to erect monuments paying homage to Jesus Christ the Redeemer on the mountain tops of the Italian peninsula (Cuaz 2009, pp. 62–63). Based on Gaspari's reconstruction (2021), it can be argued that both the conception and construction of these monuments were an "emergent effect," where the ecclesiastical authorities of the Vatican were partially interpreted by the local churches, especially by the mountain people seeking redemption (Gaspari 2021, p. 43). An International Roman Committee for the Solemn Homage to Jesus the Redeemer called for the erection of Homage to the Redeemer "to celebrate the nineteen centuries of Christianity and as a symbol of Italy's consecration to Christ" (Morettoni n.d., Cathopedia). However, the nature of the request seems much more modest compared with the actual results, which included the erection of approximately twenty religious artefacts, including monuments such as crosses, chapels, statues of Christ and an unrealised Madonna of the Snow in Emilia-Romagna. The Jubilee Year 2000 also seemed like a propitious time for the erection of new crosses or the maintenance of existing ones[1]. To the best of my knowledge, there is limited evidence to support this claim, except to note that the work of both Löwer (2019) and Millesimi (2022) points to a significant number of crosses erected in the Alps and the Apennines after the year 2000.

Mathieu (2006) concluded his essay on the sacralisation of European mountains by stating that in the more recent past, they have acquired new significance, making the weight of this sacredness more uncertain, especially when compared with other continents. Cuaz (2006) explains that erecting crosses and conducting rituals at high altitudes actually Christianised the peaks, reaffirming "in the face of the modern world [...] the role of the Church and of priestly intermediation, which had to be present in every sphere touched by man" (p. 63). In this regard, the anthropologist, Gallini (2007), aptly noted how the erection of a cross is a "strong institutional hold on a space transformed into Christian territory" (p. 50). The fact of erecting the crosses on state or municipal territories and the traditional presence of civil and religious authorities during their inaugurations were then the expression of a "synergy of two orders [the religious and the civil] united by a symbolic indistinction" (Gallini 2007, p. 51) that today has—in fact—separated. Chronologically tracing the events and reasons for this fracture in the 21st century is the aim of the next section.

## 5. The Contestation of Crosses on Summits

The mobilisation against the presence of crosses on the mountains of the European continent has manifested itself over the past decades in at least two different ways. First, there have been repeated acts of vandalism in Italy, Spain, Germany and on the Swiss–French border[2]. Second, one can observe in Italy the periodic manifestation of a movement

of opinion, mainly in the media, in favour or against the erection of new crosses that has, in fact, originated from the mountaineering sphere. The vandalism perpetrated on crosses and other signs of the sacred is certainly a practical act of the individual's denial of the value of a worldview. Such acts can be met with either censure or applause from public opinion (de Certeau and Hameline 1978). However, of greater interest are contestations involving public opinion because they allow the observation of a collective phenomenon through the analysis of the genesis of the discourse where social reasons can be grasped. These are mainly related to the pluralistic context and the view of the mountain as a place of leisure.

*5.1. The Origins of Religious Pluralism*

The origin of this debate in the mountaineering sphere can be seen in the positions of the Italian Alpine Club (Club Alpino Italiano, hereinafter CAI), which has been a protagonist of reflection on the subject on several occasions. In 1997, the Ligurian Piedmontese Valdostan Scientific Committee of the CAI focused on the signs of popular religiosity in the Western Alps. In the accounts of the relationship between humans and mountains, we find in the introduction that a "world, sometimes dormant under the influence of scientism or rejected in the effort of alpine performance," resurfaces (Smiraglia 1997, p. 6). Alongside the increasing number of scientific and tourist publications discussing the signs of the sacred in the mountains, the theme gains recognition and generates interest in both the tourist market and the local community's social re-appropriation of memories.

The thoughtful reflections primarily involving scholars prompted a debate that began in the late summer of 2005 with an event. This event sparked the anti-crucifix movement in Italy and received significant media coverage. Four alpine guides from Valtellina carried a ceramic statue of Buddha, weighing about 20 kg and 1.30 m in height, up Pizzo Badile, which rises 3308 m in Valmasino in the province of Sondrio (Corvi 2005). The incident had a considerable impact, reflected in blogs and channels dedicated to the mountains, to the extent that a video was created a couple of years later, documenting the action and relaying reactions from various mountain blogs[3]. After the Buddha was found in pieces on a ledge a few years later, director Alberto Valtellina created a video named "Iconoclasti" (2013), which recounts the feat as part of an artistic performance that culminated in the restoration of the Buddha in Brera, following the logic of artistic exhibition[4]. This occurrence, in itself, is not isolated; in subsequent years, one can find similar emulators on other Italian Alpine northeastern peaks, such as the *Dinosaur* on Mount Pelmo[5] or the *Sleeping Madonna* on Sass da Ciamp[6].

The sociological interest stems from the fact that this feat fostered a discussion that had begun earlier on the website of the Union of Agnostics and Rationalist Atheists (Unione degli Atei e degli Agnostici Razionalisti, hereinafter UAAR), where the writer and mountaineer, questioned the "appropriateness and meaning of the installation of crosses, various artefacts, and other unambiguously religious symbols on mountain peaks" (Rota 2005).

In January 2006, a conference was organised by the Luigi Bombardieri Foundation, in collaboration with the Sondrio CAI, entitled *I segni del sacro sulle montagne. Immagini e opinioni*. The conference aimed to reflect on all the representations of the sacred, hierophanies and signs. A report on this event concludes, "Differences must therefore be understood between the genuine meaning of symbols of sacredness as expressions of a religiosity experienced in the mountain pastures and inhabited valleys, as reasons for existence and moments of life, and the extraneousness of the symbols on the peaks, where there has never been a lived life, but only an occasional tourist visits" (CAI 2006). The conclusions recognise the legitimacy of lived religion for communities, partially disqualifying the symbols on summits by attributing them to leisure visits. Interestingly, even the conquest of the summit, which previously symbolised a synergy between the secular and the religious through the cross, is now attributed to a purely secular activity that diminishes the pedagogical value

of mountaineering ascent, as Simmel (1991) stated in his short article on Alpine journeys in 1895.

### 5.2. Mountain Wilderness and Sobriety for Mountains

In 2013, a second wave of media mobilisation against crosses emerged when the Italian section of the environmental association, Mountain Wilderness (hereinafter MW), with the support of other environmental associations, launched an appeal to public bodies against "infrastructures and crosses on the peaks" to "regulate, with respect for the places and the different sensitivities of their visitors" (Mountain Wilderness 2013). The goal was to oppose the placement of bulky and impactful artefacts, including religious symbols that ostentatiously represent religiosity, whether within or outside a tourism context that also uses the sense of the sacred. This stance echoes initiatives that primarily question the compatibility of such signs with the naturalness of areas around mountain summits. The triggering fact was the case of the Trekking of the Thinking Christ of the Dolomites in the Paneveggio and Pale di San Martino Park in eastern Trentino. This case posed a significant challenge to the local church, which, through the Delegate for the Pastoral Care of Tourism, consistently called for restraint. The placement of Christ and a cross in such a natural environment creates a clear dissonance between environment, tourist experience and religious meanings. According to the local church, the criteria for placing crosses or other sacred signs in mountainous territories involve actively listening to the message of the mountain, embracing natural spontaneity, and engaging in discussions with the local communities concerned (Interviews 24 and 25 February 2013).

This call by religious institutions can be understood within a broader process that addresses the responsibility of Catholicism towards nature, which is a novelty in the social doctrine of the Church. Following the Second Vatican Council, the cause of environmental protection and the preservation of creation gained momentum, incorporating the protection of mountains into the Church's social doctrine. The alignment of the Archdiocese of Trento's Pastoral Care of Tourism with secular environmentalism is noteworthy. It confirms Turina's (2022) observation regarding the innovative elements that were first deliberated in the previous pontiff's consideration of ecological crisis due to excess consumption, and then introduced in Pope Francis' encyclical letter, *Laudato si'* (Francis 2015). In terms of balancing legitimacy, the *Laudato si'* "marks the inclusion of theological perspectives, as well as social, cultural, and economic perspectives, into the official discourse of the Church's central authority, which had previously developed on the periphery of that authority" (Turina 2022, pp. 32–33). While this peripheral openness mainly refers to relations with non-European Catholicism, the commitment to nature conservation (exemplified by the case of the Thinking Christ) indicates similar consequences. This entails an opening of the Catholic Church to a broader public and, certainly, a greater openness to interreligious relations (Turina 2022). Moreover, it implies an implicit revision of the modernist and anthropocentric position contained in Leo XIII's encyclical letter, *Rerum Novarumi*, in 1891 (Hart 2006).

### 5.3. An Astylar Cross on Mount Baldo

The third and most recent wave of mobilisation against crosses on mountains dates back to 2021–2022[7]. The triggering event was the initiative of an entrepreneur from Verona whose family has consistently supported sports activities aimed at promoting the individual and inspired by Christian values (Interview 2 October 2019). The entrepreneur acquired an 18-metre-high astylar cross, created by a contemporary Roman artist, depicting Pope John Paul II, now a saint, leaning towards the crucified Christ[8]. The plan was to erect this cross on Mount Baldo, located on the eastern ridge of Lake Garda, upwards of the Malcesine–Monte Baldo cable car, an area of tourist and naturalistic value. To proceed, a concession for the use of a portion of land was requested and obtained from the municipality of Malcesine in autumn 2020. However, the agreement with the municipality faced opposition from members of the municipal council, sparking mobilisation through various channels,

including media, legal action and citizen awareness. Municipal councillors, along with numerous environmental and mountaineering associations, launched an online petition[9] to local, regional and national political decision-makers, demanding the halt of the installation project. Thanks to a massive relaunch in both the generalist and specialised media on mountain issues, the petition garnered approximately 28,000 signatures in just two months, with around one tenth of the signatories leaving comments.

In response, a committee named *Comitato Amiche e Amici del Baldo* was formed in the concerned municipality. This nonpartisan group of citizens aimed to promote and protect the environmental integrity of the Monte Baldo area[10]. The Committee initially addressed the mayor, urging him to abandon the project. The significance of their message was highlighted by the explicit reference to Pope Francis' encyclical letter, *Laudato si'*, on the environment as a collective good (Letter, 31 March 2021). Subsequently, the Committee called for referendum rallies to address the question of the cross installation (Letter 26 August 2021). Simultaneously, a fundraiser was launched to support the legal costs for an appeal initially pursued by the Mountain Wilderness Association, which was later downgraded to an appeal to the Regional Administrative Court. In September 2021, a demonstration took place at the proposed cross site, with approximately a hundred participants. Additionally, parallel to the opposition to the astylar cross, collective action supported the initiative to have Monte Baldo registered on the UNESCO World Natural Heritage List. This initiative led to the establishment of the social promotion association, "Monte Baldo World Heritage Site" in March 2021 (Guarelli and Peretti 2021).

In summary, local stakeholders raised concerns about the cross's installation on two fronts. Firstly, they highlighted the ecological consequences of increased tourism resulting from the presence of the cross. Secondly, they questioned the true intention behind the initiative, interpreting it as a profit-oriented venture without any positive impact on the local community. The nationwide mobilisation reignited a protest that, as discussed earlier, revolved around opposition to additional crosses and religious symbols in natural settings and concerns about the loss of naturalness and ecological damage due to the potential influx of tourists, especially considering the recent surge in mountain tourism in Italy during the pandemic period (Macchiavelli 2022).

The placement of John Paul II's astylar cross on the slopes of Mount Baldo, in an area owned by the local church and near an important shrine dedicated to the Virgin Mary, has generated significant applause from both detractors and supporters of the cross. Both sides consider themselves victorious.

The resolution of this particular conflict serves as an example of the symbolic division between the secular and religious domains and the loss of the cross's "form and function of cosmic protection" (Gallini 2007, pp. 50–51). Nevertheless, the authority of the Church continues to be acknowledged within its territory, albeit partially confined to the sphere of private property.

## 6. Discussion and Conclusions

The symbolism of the cross on mountain peaks retains its significance, not only as an aesthetically appreciated element within the distinctive landscape that it occupies, but increasingly as a subject of social and normative debate, becoming a cultural point of contention. The reflection on summit crosses evokes controversy and division within the social sphere, involving public opinion, social institutions, associations, scholars and intellectuals at various levels, both nationally and internationally. The debate oscillates between acknowledging the universal value of this symbol and advocating secularism, even in open spaces such as mountain summits. However, even today, the expressive power of the symbol does not easily conform to a single interpretation. The media context in which the cross has become a subject of controversy is eminently secular, despite the pluralistic positions expressed by secular or atheist environmental activism (see Section 5.1) and by religious authorities that recognise love for the mountains as a source of introspection and pastoral commitment (see Sections 4 and 5.2).

In 1931, Guénon (1957) asserted that the cross was a universal symbol and that Christianity, at least in its outward appearance and familiarity to most, had lost some of its symbolic character, reducing it to a historical fact (pp. 5–6). It should be noted with caution that while crosses still possess universality, the previous emphasis on the historical fact, specifically the life and death of Jesus Christ, is increasingly overshadowed by political and identity appropriations and reinterpretations, as evidenced in the sociological literature and as discussed in this study. When Joppke (2018) compared the European High Court's stance on crucifixes and veils, he noted that "the crucifix, even the barren Latin cross that does not exhibit the body of Christ, is intrinsically religious, being a core element of Christian liturgy" (p. 242). In the media context where these social actions took place, nowadays, the cross is primarily considered an artefact that interacts with the landscape and the mountain. In other words, both belong to the same order of reality, and transcendence is no longer "leading the spiritual eye upwards, where, above that which is attainable only with the greatest danger, there lies that which no more power of will can reach" (Simmel 1993, p. 181).

To comprehend the actual status of summit crosses, Simmel's (1998) sociological analysis of the fundamental qualities of the spatial form remains relevant. In the discursive context under examination, the summit cross can be interpreted as "the symbolic expression of the 'centre of rotation': the spatial fixity of an object of interest [which] produces certain forms of relations that cluster around it" (p. 537). This role held true historically and socially during the period of Christianization of mountain peaks. However, the centre of rotation persists today in the social use of summit crosses, serving as a focal point for both religious and secular mountaineers, vacationers seeking amusement and individuals inspired by the inherent spirituality of mountain climbing. These perspectives continue to polarise public opinion, while acknowledging the religious–secular symbolic distinction that has become apparent among the various social actors involved in the controversies discussed in this study.

The Italian contestation of crosses encompasses all three modes of culturised religion noted by Astor and Mayrl (2020): constituted culture, pragmatic culture and identity. The opposition to crosses emerges within the context of the diminishing symbolic dominance of Catholicism. It highlights how a religious artefact can influence religious sentiments, such as in the Christianisation of mountain peaks, and underscores the distinction between a religion lived by mountain communities and a religion merely symbolised by cultural objects. This mobilisation not only amplifies non-religious or anti-religious sensibilities, as seen in the campaigns by the UAAR and the MW, but also stimulates efforts towards a deeper scientific understanding of the presence of crosses. Millesimi (2022) particularly hopes that the transdisciplinary study of summit crosses can make them "useful" for a greater comprehension of the climatic and natural conditions of mountains. The symbolic dimension is inherent to human nature, enhancing its expressive potential; dismissing its usefulness on epistemological grounds is unfounded (Ricoeur 1969). However, when a symbol such as the summit cross becomes a cultural object, its status is determined through analytical decisions made by observers. The critical observers examined in this study ascribe indirect economic value tied to tourism, as well as political and identity value, to summit crosses (Griswold 2013). Cultural objects possess the ability to be produced, reproduced, exchanged, sold, bought and consumed in both real and symbolic markets.

Regarding the question of whether the cross on mountains is a passive religious symbol, the answer appears to be only partially negative. When the debate on the display of the crucifix (a specific instance of embodiment) reached various levels of the European Court of Human Rights, the value attributed to the religious symbol shifted. Breskaya et al. (2022) analyse this shift, observing that the cross acquired value not solely based on its symbolic efficacy, but also due to its deterrent effect. The pronouncements on the display of the crucifix demonstrate the complex relationship between what is considered appropriate for religion and culture, and the contexts in which these rules can be correctly applied. Breskaya et al. (2022) accept Beaman's (2012) still valid framing, and acknowledge

not only the symbolic value but also the power and efficacy of the crucifix. Paradoxically, crosses on mountains have garnered greater significance and have had a deterrent effect on secular or non-believing individuals who oppose the installation of further crosses, yet are more inclined to preserve existing ones. Burchardt (2020) effectively explores this issue and discusses the passive role of this symbol, considering the "active and, in fact, agentive nature" of religion as heritage that "involves commitments and practices that people mobilise for valued ends" (p. 159). The values driving the contestation of the crosses on mountain peaks revolve around earthly concerns about the expression of diversity, and to a lesser extent, the climate crisis and safety in the mountains. Moreover, for "secular sections of the population, heritage religion offers ways to affirm [...] the unique nexus between their religious tradition and national history and identity" (Burchardt 2020, p. 196). This interpretation of the cross's identity is expressed both on a personal and collective level, amplified through media platforms with global resonance. However, the conclusion of the matter of the astylar cross on Mount Baldo (Section 5.3), where both sides consider themselves victorious and satisfied, demonstrates the continuous negotiation of boundaries among the non-binary social relationships of the secular and religious, as defended by Knott (2013).

In an old exchange involving a French theologian, a philosopher and a Dominican pastoralist regarding the preservation of sacred objects, it was strongly emphasised that "the affection or disaffection of Christians for their objects received from the past is quite another thing [...] than a simple question of culture or inculturation, of good or bad taste" (de Certeau and Hameline 1978, p. 11). The relationship between preservation and vandalism, however, alludes to two ideas that are still operative today: the aestheticisation of the past and ownership as well as the Church's need for self-reflection and the message it intends to convey about God (de Certeau and Hameline 1978, p. 15). This message is also the medium: a wooden, stone or metal cross, both in the present and in the past, but perhaps not in the future.

**Funding:** This article was written during the project "Il turismo degli itinerari e dei cammini: una ricerca qualitativa sull'uso sociale delle eredità storico-religiose" funded by the Department of Human Sciences, University of Verona, Italy (February 2023–January 2024).

**Institutional Review Board Statement:** The study was conducted in accordance with the Declaration of Helsinki.

**Informed Consent Statement:** All participants were informed, and no risks were associated with their participation.

**Data Availability Statement:** This study did not produce datasets because it is an ethnographic research.

**Acknowledgments:** The author thanks the Editors of this Issue Olga Breskaya and Giuseppe Giordan for their invitation. The author thanks her scientific responsible Sandro Stanzani for the support.

**Conflicts of Interest:** The author declares no conflict of interest.

## Notes

[1]　Testimony of the Head of the Tourism Pastoral of the Archdiocese of Trento (Interview, 24–25/02/2011).

[2]　This is not the place for a repertoire, but crosses in Switzerland, Bavaria and on the border between France and Spain have been vandalised. A mountain guide in Switzerland took responsibility for such vandalism and was condemned: cf. Roulet (2012).

[3]　A Buddha on the Badile, in Masino Climbing https://www.youtube.com/watch?v=yvbh1uD-Mtw (accessed on 23 February 2023).

[4]　*Iconoclasti, il sorriso del Buddha* di Alberto Valtellina (Italia 2013, 7′): https://www.albertovaltellina.it/portfolio/iconoclasti/ (accessed on 29 May 2023).

[5]　https://www.valdizoldo.net/it/attivita/culturattiva/la-macchina-del-tempo/monte-pelmo-dove-osano-i-dinosauri (accessed on 29 May 2023).

[6]　https://www.gulliver.it/itinerari/toac-monte-da-medil-anello-per-cima-da-ciamp-e-sas-da-ciamp/2016/12/28/236163/#&gid=1&pid=6 (accessed on 29 May 2023).

7     The materials in this section summarise an ethnography that made use of non-directive interviews, informal conversations, the collection of secondary materials and a netnography.

8     https://www.youtube.com/watch?v=DLYNGdx3PLY (accessed on 20 March 2023).

9     https://www.change.org/p/una-croce-di-18-metri-sul-monte-baldo-giardino-botanico-d-europa?redirect=false (accessed on 23 April 2023).

10    https://amicheamicidelbaldo.wordpress.com/about/ (accessed on 30 May 2023).

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
