# Peer review of "Religion upon the Mountains: From Christianisation to Social Actions against Summit Crosses in Italy"

_religions, doi:10.3390/rel14081056_

Round 1

Reviewer 1 Report

1.The paper of the author does really contribute to a better understanding of the public debates regarding the presence of crosses and crucifies in public space in Italy from the sociological point of view, and in some extent from an historical perspective, but not from the legal and theological point of view, as it was expected.

2. When somebody speaks about a monotheistic religion, like Christianism and its faith, he has first of all to examine the main theological and legal texts, and then to appeal to the text of different authors (theologians and jurists). Otherwise, the author risk to present only a sociological point of view or his own opinions about the presence of crosses and crucifies in the public space and their impact in our social- media.

Therefore, I propose to the author to take seriously into consideration these collegial suggestions, in order to give to his paper a better documentary basis, and not to remain tributary only to the literature of speciality, since only in this way we could prove a real engagement with sources.

Author Response

Article title: Religion upon the Mountains: From Christianisation to Social Actions Against Summit Crosses in Italy

Dear Reviewer,

I would like to thank you for the careful reading of my article. I have to apologise to have been so long in submitting the revised text but it is necessary to inform you about a fact. When I received the reviewers’ comments a new great debate literally exploded in the Italian media (the last ten days of June and the first week of July) about the mountaintop crosses. I took the time to follow it before revising this article.

Finally, as I am a qualitative and abduction-driven researcher, I concluded that it was necessary for me to let that debate “deposit” a little bit. You will not find any reference to the last debate and tried to stay on the purpose of the paper as it was initially conceived and simply take your precious suggestions to improve the article. In any case, I consider that time of observation essential to understand if the article here re-submitted is touching the core of the revived debate.

Here follows my punctual reply.

Sincerely,

Author

REVIEWER 1 – ROUND 1

1.The paper of the author does really contribute to a better understanding of the public debates regarding the presence of crosses and crucifies in public space in Italy from the sociological point of view, and in some extent from an historical perspective, but not from the legal and theological point of view, as it was expected.

It is true: I considered and favoured historical and sociological sources but still in a rough way the legal point of view. I tried to consider this one as being cautious to keep the purpose of the paper, as the materials analysed are public and political debates not actually directly impacting the legal context, as I did not find any particular lawsuit in this area in Italy (in France there is one).

In my opinion, it has been useful to inform the reader how the new crosses must be managed from a legal point of view: it relates to the landscape aesthetics and anthropic impact of new installations on mountaintops. Collected materials question this point and the conclusions could be translated into possible implications for policy makers (if the article would have aimed, or will aim at some point, at these practical consequences).

From the theological point of view, I don’t really feel very comfortable with such primary sources: I have very limited knowledge in this domain and risk being less informative than expected. Nevertheless, I tried to embrace your suggestion and consider the vast debate on spiritual and environmental ethics from the borders and within Christianism. Here I feel more comfortable from a scientific point of view.

2. When somebody speaks about a monotheistic religion, like Christianism and its faith, he has first of all to examine the main theological and legal texts, and then to appeal to the text of different authors (theologians and jurists). Otherwise, the author risk to present only a sociological point of view or his own opinions about the presence of crosses and crucifies in the public space and their impact in our social- media.

I perfectly understand the point and recognise the great importance of texts. I am not enough erudite to confront with the “Christology of the Cross” or the likewise vast “Theology of the Cross” from that point of view (even if I read a pair of article not cited here like this and this). I have been educated and trained in human sciences, even if I have a superficial education in history of religious arts. The reason of privileging sociological point of view is primary connected to the materials collected: public debate is at low level of theological implications. Even if I have some knowledge of the Social Doctrine of the Church (now explicitly cited), I would prefer to refer to the works of a scholar who has extensively studied it from a sociological point of view (see in the text Turina).

Therefore, I propose to the author to take seriously into consideration these collegial suggestions, in order to give to his paper a better documentary basis, and not to remain tributary only to the literature of speciality, since only in this way we could prove a real engagement with sources.

The text was revised as follows:

-          In paragraph 3 was added one subparagraph specifically developing the matter of religious studies and environmental humanities from an ethical point of view.

-          Paragraph 5 furtherly develops the discussion by engaging with Authors who explored the legal side of symbols (cf. Joppke and Burchardt).

From the original text of about 7750 characters, the article is now about 9000.

Reviewer 2 Report

Dear Author,

I would like to congratulate you on your choice of topic and theme. In the context of contemporary cultural changes, the undertaken research project seems both important and necessary. This article identifies a distinctive configuration that has emerged within the public discourse concerning the mountain environment and the objects placed there, forming.

The Author correctly indicated the research problem, purpose, and method. This study is well designed, is written in a clear, adequate language and provides interesting reading. No doubt, that it has a good level. However, in my view, it needs improvements before it can be published. However, I do not see the context being properly presented. An unfamiliar reader may have a big problem in this respect. Hence, I would expect the article to be supplemented with the local context.

The language is correct and understandable, but there are editing errors.

Author Response

Article title: Religion upon the Mountains: From Christianisation to Social Actions Against Summit Crosses in Italy

Dear Reviewer,

I would like to thank you for the careful reading of my article. I have to apologise to have been so long in submitting the revised text but it is necessary to inform you about a fact. When I received the reviewers’ comments a new great debate literally exploded in the Italian media (the last ten days of June and the first week of July) about the mountaintop crosses. I took the time to follow it before revising this article.

Finally, as I am a qualitative and abduction-driven researcher, I concluded that it was necessary for me to let that debate “deposit” a little bit. You will not find any reference to the last debate and tried to stay on the purpose of the paper as it was initially conceived and simply take your precious suggestions to improve the article. In any case, I consider that time of observation essential to understand if the article here re-submitted is touching the core of the revived debate.

Here follows my punctual reply.

Sincerely,

Author

REVIEWER 2 – ROUND 1

I would like to congratulate you on your choice of topic and theme. In the context of contemporary cultural changes, the undertaken research project seems both important and necessary. This article identifies a distinctive configuration that has emerged within the public discourse concerning the mountain environment and the objects placed there, forming.

Many thanks for the reading and the appreciation.

The Author correctly indicated the research problem, purpose, and method. This study is well designed, is written in a clear, adequate language and provides interesting reading. No doubt, that it has a good level. However, in my view, it needs improvements before it can be published. However, I do not see the context being properly presented. An unfamiliar reader may have a big problem in this respect. Hence, I would expect the article to be supplemented with the local context.

I engaged with the local context, adding some punctual considerations about the Alps and their specific and crucial role of “borders”.

Then I explained the specificity of Italian regulation of landscape: in my opinion, it is useful to inform the reader how the new crosses must be managed from a legal point of view. It relates to the landscape aesthetics and anthropic impact of new installations on mountaintops. Collected materials question this point and the conclusions could be translated into possible implications for policy makers (if the article would have aimed or will aim in a second time at these practical consequences).

Adding a paragraph on environmental, religious and spiritual ethics, I wanted to give a more detailed context into the religions-law-environment nexus.

The text was revised as follows:

-          In paragraph 3 was added one subparagraph specifically developing the matter of religious studies and environmental humanities from an ethical point of view.

-          Paragraph 5 furtherly develops the discussion by engaging with Authors who explored the legal side of symbols (cf. Joppke and Burchardt).

From the original text of about 7750 characters, the article is now about 9000.
